# Optimization of Ammonia Stripping of Piggery Biogas Slurry by Response Surface Methodology

**DOI:** 10.3390/ijerph16203819

**Published:** 2019-10-10

**Authors:** Mengyuan Zou, Hongmin Dong, Zhiping Zhu, Yuanhang Zhan

**Affiliations:** Institute of Environment and Sustainable Development in Agriculture, Chinese Academy of Agricultural Sciences, Beijing 100081, China; condor1228@sina.cn (M.Z.); donghongmin@caas.cn (H.D.); zzyh727@126.com (Y.Z.)

**Keywords:** response surface methodology, biogas slurry, ammonia stripping

## Abstract

Ammonia stripping is a pretreatment method for piggery biogas slurry, and the effectiveness of the method is affected by many factors. Based on the results of single-factor experiments, response surface methodology is adopted to establish a quadratic polynomial mathematical model relating stripping time, pH value and gas flow rate to the average removal rate of ammonia nitrogen to explore the interactions among various influencing factors, obtain optimized combined parameters for ammonia stripping, and carry out experimental verification of the parameters. The results show that when hollow polyhedral packing is adopted under operating conditions including a stripping time of 90 min, pH value of 11, gas flow rate of 28 m^3^/h, gas–liquid ratio of 2000 and temperature of 30 °C, the average removal rate of ammonia nitrogen in biogas slurry can reach approximately 73%. The experimental value is only 4.2% different from the predicted value, which indicates that analysis on the interaction among factors influencing ammonia stripping of biogas slurry and parameter optimization of the regression model are accurate and effective.

## 1. Introduction

With a high ammonia nitrogen concentration, low carbon-to-nitrogen ratio and poor biodegradability, biogas slurry is a kind of high-concentration organic wastewater with a complex composition of pollutants [1]. Biogas slurry has few available nutrients. Slurry released via direct mass discharge is difficult to completely absorb in situ and may cause serious pollution of farmland environments. Biogas slurry requires pretreatment or in-depth combination treatment. The stripping method utilizes the difference between the actual concentration of ammonia nitrogen contained in wastewater and the equilibrium concentration by providing full contact between gas and water under alkaline conditions to remove ammonia nitrogen [2]. The influencing factors of nitrogen removal efficiency include the gas–liquid ratio, temperature, stripping time and pH [3,4]. Because of its high denitrification percentage, flexible operation and minimal land use, ammonia stripping is widely used in the pretreatment of various types of high-ammonia nitrogen wastewater, including landfill leachate [5,6], municipal wastewater [7], aquaculture wastewater [8,9] and biogas slurry [2,10,11,12].

Response surface methodology is a design optimization method that combines mathematical methods and statistical analysis and is based on the visual analysis of nonlinear multivariable data and multiple regressions [13,14]. Response surface methodology has relatively few test groups and the test cycle is usually short, thus manpower and material resources are saved; the precision of the regression equation is high and the predictability is good. Response surface methodology can be used to study interactions among influencing factors and enables intuitive judgments of optimal values by graph analysis. Based on the above characteristics, response surface methodology has been applied in many fields, including extraction processes [15,16], material preparation [17,18] and wastewater treatment [19,20,21]. Box–Behnken design is a response surface design method that was proposed by Box and Behnken and is often used for two to five test-factors. The design assumes the existence of quadratic terms for factors within the test range, and the test points are the midpoint and central point of each edge of a coded cube; furthermore, the operating cost is lower than that of central composite design with the same number of factors [14,22]. Based on single-factor experiment, this study optimizes the technological parameters for ammonia stripping of piggery biogas slurry through Box–Behnken design in response surface methodology.

## 2. Test Material, Apparatus and Method

### 2.1. Test Material

The biogas slurry used in the test was wastewater generated from anaerobic fermentation in a large-scale piggery biogas project in Hengshui City, Hebei Province. The mass concentration of ammonia nitrogen was 650 to 710 mg/L after flocculation, solid–liquid separation and 50 μm tape filtration pretreatment. The packing used in the test consisted of hollow polyhedral spheres made of polypropylene. The specific surface area was 500 m^2^/m^3^, the voidage was 84%, the dry packing factor was 844 m^−1^ and the density was 85,000 pcs/m^3^. The absorption liquid used in the test was 1 mol/L sulfuric acid solution. The neutralizing liquid used in the test was 0.5 mol/L sodium hydroxide solution.

### 2.2. Test Apparatus

As shown in Figure 1, the test apparatus includes a control unit and stripping unit. The stripping unit included a stripping tower, liquid storage tank and absorption apparatus. The control unit was adjusted through switches to set the frequencies (powers) of the PLC-controlled aeration fan, stirrer, heating rod, and biogas slurry and alkaline solution peristaltic pumps. The material of the stripping tower was polymethyl methacrylate, the height was 1.5 m, and the inner diameter was 0.15 m (the height–diameter ratio was 10:1). The material of the liquid storage tank was stainless steel, and the effective volume was 150 L. The materials of the absorption tank and neutralization tank were polymethyl methacrylate, and the effective volume was 5 L.

When stripping started, the aeration fan provided aeration, and the air in the stripping tower flowed upward while the biogas slurry was transported by the peristaltic pump and discharged downward. The slurry entered the liquid storage tank after reaching the bottom of the stripping tower, and then the biogas slurry peristaltic pump removed more slurry and provided circulation. After the air and stripping offgas were discharged from the exhaust port, ammonia gas was absorbed by the absorption liquid and the offgas was discharged into the atmosphere through NaOH solution to neutralize acidity. Control units were used to control three sets of parallel discharge stripping units and were used in parallel tests.

### 2.3. Test Method

In our research group, Sui showed that the removal rate of ammonia nitrogen in biogas slurry was 81.84% under the operating conditions of pH 10.5, gas flow rate 5 m^3^/h, gas–liquid ratio 2000–2500 and temperature 30 °C [23]. However, the stripping time was more than 12 h. Therefore, this test was based on the parameters of pH 10.5, gas–liquid ratio 2000 and temperature 30 °C. The pH value and gas flow rate were appropriately increased.

As shown in Table 1, the single-factor influences of stripping time, pH value and gas flow rate on the ammonia stripping effect of biogas slurry were explored first. The stripping tower was filled with hollow polyhedral packing material to 1 m height, and 20 L pretreated piggery biogas slurry was added to the liquid storage tank. The stirrer was turned on (80 to 120 r/min), the frequency of the aeration fan was set, and the biogas slurry peristaltic pump was adjusted accordingly. The gas–liquid ratio was kept at 2000, and the biogas slurry in the liquid storage tank was heated to 30 °C. Stripping was started after adding NaOH solution to the liquid storage tank via the alkaline solution peristaltic pump to adjust the pH of the biogas slurry to a stable value, and average ammonia nitrogen removal rate was measured after stripping.

The ammonia nitrogen concentration was measured by salicylic acid–sodium hypochlorite spectrophotometry. The test instrument was HACH COD Reactor Model DR 6000 (HACH Company, USA). The average ammonia nitrogen removal rate (Y) was calculated by the following equation:(1)$$Y=\frac{\left(C-C^{\prime }\right)}{C}*100\%$$
where C is the ammonia nitrogen concentration in the test water before stripping and C’ is the ammonia nitrogen concentration in the test water after stripping.

Based on the results of the single-factor experiment, Design-Expert 8.0.6.1 software was used for the Box–Behnken design. Three influencing factors, i.e., stripping time, pH value and gas flow rate, are represented by A, B and C, respectively. Low, medium and high levels of the three independent variables are coded with −1, 0 and 1, respectively, and the ammonia nitrogen removal rate was taken as the response value Y to obtain a test program including 15 test points for three levels of the three factors. By establishing a quadratic polynomial regression model and drawing a contour plot and response surface plot for each pair of factors, the interactions among the influencing factors were analyzed, and optimized combined parameters are obtained. The factors and levels of the response surface experiment are shown in Table 2.

## 3. Results and Discussion

### 3.1. Influence of Stripping Time on the Effect of Ammonia Stripping

Under an initial pH value of 10.5 and gas flow rate of 20 m^3^/h, biogas slurry stripping times including 15, 30, 45, 60, 75, 90, 105 and 120 min are applied. The test results are shown in Figure 2. The average concentration of ammonia nitrogen decreases with increasing stripping time, while the average removal rate increases with increasing stripping time. After stripping for 120 min, the average concentration of ammonia nitrogen falls to 189.3 ± 28.1 mg/L from 566.0 ± 16.0 mg/L, and the average removal rate is 66.5%. After stripping for 30 min, the average concentration of ammonia nitrogen falls to 258.0 ± 22.0 mg/L, and the average removal rate is 54.4%. When the stripping time is further prolonged to 120 min, the average removal rate rises by only 12.1%. At stripping times of 45 min or longer, the average concentrations of ammonia nitrogen have no significant difference (*p* > 0.05). Under the above test conditions, both the effectiveness and economic efficiency of ammonia stripping are optimized at a stripping time of approximately 30 min.

### 3.2. Influence of pH Value on the Effect of Ammonia Stripping

Under a gas flow rate of 20 m^3^/h, the pH value of the biogas slurry is adjusted to 11, 11.5 and 12. The results in Figure 3 show that after stripping for 120 min, at pH values are 11, 11.5 and 12, the average concentrations of ammonia nitrogen fall to 213.3 ± 33.6, 204.0 ± 20.4 and 138.0 ± 2.0 mg/L from 681.3 ± 16.7, 696.7 ± 56.4 and 526.0 ± 52.6 mg/L, respectively, and the average removal rates are 68.7%, 70.7% and 73.8%. Under the same stripping time, the average ammonia nitrogen removal rate increases with increasing pH, but at pH values of 11, 11.5 and 12, there is little change in the average removal rate. At pH values ranging from 11 to 12, there is no significant difference in the removal effect of ammonia nitrogen among different pH values (*p* > 0.05).

### 3.3. Influence of Gas Flow Rate on the Effect of Ammonia Stripping

At a pH value of 10.5, the gas flow rates are adjusted to 20, 24 and 28 m^3^/h. The results in Figure 4 show that after stripping for 120 min, at gas flow rates of 20, 24 and 28 m^3^/h, the average concentrations of ammonia nitrogen fall to 189.3 ± 28.1, 150.7 ± 25.0 and 182.0 ± 25.5 mg/L from 566.0 ± 16.0, 599.3 ± 25.8 and 578.7 ± 59.9 mg/L respectively, and the average removal rates are 66.5%, 74.9% and 68.5%. Under the same stripping time, the average ammonia nitrogen removal rate changes with gas flow rate in the order 24 > 28 > 20 m^3^/h, but when gas flow rates are 20, 24 and 28 m^3^/h, there is no significant difference in the average removal rate (*p* > 0.05). 

### 3.4. Response Surface Analysis Results

The response surface experiment plan and results are shown in Table 3. Design-Expert 8.0.6.1 software is used to conduct fitting analysis via multiple linear regression and nonlinear (quadratic polynomial) regression of the test data. The multiple linear regression equation relating the ammonia nitrogen removal rate (Y) to the stripping time (A), pH value (B) and gas flow rate (C) is Y = 69.00 + 3.06A + 2.89B + 5.97C, the determination coefficient is R^2^ = 0.8376, the adjusted determination coefficient is R^2^ = 0.7932 and the root-mean-square error (RMSE) is 11.92; the quadratic polynomial regression equation relating the ammonia nitrogen removal rate (Y) to the stripping time (A), pH value (B) and gas flow rate (C) is Y = 69.74 + 3.06A + 2.89B + 5.97C − 0.29AB + 0.56AC − 2.79BC − 0.76A^2^ + 1.55B^2^ − 2.17C^2^, the determination coefficient is R^2^ = 0.9613, the adjusted determination coefficient is R^2^ = 0.8918 and the RMSE is 7.38. The determination coefficient R^2^ represents the model fitting degree. As seen from the determination coefficients of the multiple linear regression equation and quadratic polynomial regression equation, the fitting degree of the multiple linear regression equation is not good, and the model predictability is poor. The root-mean-square error (RMSE) indicates the degree of dispersion of the sample, the smaller the RMSE, the more reliable the model. Therefore, the quadratic polynomial regression equation is adopted. The absolute value of each coefficient in the equation directly reflects the influence degree of the corresponding variable on the dependent variable, and positive and negative coefficients reflect the direction of the influence. As seen from the equation, within the scope of test parameters (stripping times of 30, 60 and 90 min; pH values of 11, 11.5 and 12; and gas flow rates of 20, 24 and 28 m^3^/h), the order of the effect of various factors on the ammonia nitrogen removal rate Y is C > A > B, namely, gas flow rate > stripping time > pH value.

The results of variance analysis of the quadratic polynomial regression equation are shown in Table 4. The *p-*value of the quadratic polynomial regression model is less than 0.01 and reaches an extremely significant level, indicating that the relationship between independent variables and dependent variables described in the regression model is highly significant; the *p-*values of the single terms A, B and C are all less than 0.01, indicating that the linear relationships between the single terms and the ammonia nitrogen removal rate Y are extremely significant; the *p-*value of the quadratic term BC is less than 0.05, indicating that the surface effect of this term on the ammonia nitrogen removal rate Y is significant; the *p*-values of AB, AC, A^2^, B^2^ and C^2^ are larger than 0.05, indicating that their surface effects on the ammonia nitrogen removal rate Y are not significant.

The adjusted determination coefficient of the model (R^2^) is equal to 0.8918, indicating that 89.18% of the change in response value comes from the selected variables and that the model can optimize the ammonia stripping process of biogas slurry and predict the ammonia nitrogen removal rate. Lack of fit represents the probability that the predicted value of a model does not match the actual value. The *p*-value of lack of fit is larger than 0.05, indicating that lack of fit is not significant for the model and the quadratic polynomial regression equation can be used to predict the ammonia nitrogen removal rate of pretreated piggery biogas slurry under different test conditions. 

Contour plots and response surface plots are constructed to visually reflect the influences of the interactions among the three influencing factors, i.e., stripping time, pH value and gas flow rate, on the ammonia nitrogen removal rate and to evaluate each factor under optimal conditions. Contour plots reflect the strength of the interaction between each pair of influencing factors; an ellipse represents a significant interaction, and a circle represents a nonsignificant interaction. Response surface plots reflect the sensitivity of the response value to the test parameters; a gentle slope represents that the response value is stable under variations in the test parameters, and a steep slope indicates that the response value is sensitive to changes in the test parameters.

As shown in Figure 5, the interaction between stripping time and pH value is not significant (*p* > 0.05), and the ammonia nitrogen removal rate is quite sensitive to changes in stripping time. When the stripping time and pH value are at high levels, the ammonia nitrogen removal rate is large. When the stripping time is 90 min and the pH value is approximately 12, the ammonia nitrogen removal rate is large. As shown in Figure 6, the interaction between stripping time and gas flow rate is not significant (*p* > 0.05), and the ammonia nitrogen removal rate is quite sensitive to changes in gas flow rate. When the stripping time and gas flow rate are at high levels, the ammonia nitrogen removal rate is large. When the stripping time is 90 min and the gas flow rate is approximately 28 m^3^/h, the ammonia nitrogen removal rate is large. As shown in Figure 7, the interaction between pH value and gas flow rate is significant (*p* < 0.05). When the gas flow rate is approximately 28 m^3^/h and the pH values are approximately 11 and 12, the ammonia nitrogen removal rate is large.

### 3.5. Parameter Optimization and Experimental Verification

The satisfaction function method is also known as the desirability function method. Proposed by Harrington in the 1960s, this method turns each response into an individual satisfaction value and then determines the overall satisfaction in the interval (0,1), which is convenient for quickly finding the optimal solution to a problem [24,25]. The feasibility of the satisfaction function method can be enhanced by applying the weighted geometric mean method of Derringer and Suich [26]. The optimal ammonia nitrogen removal rate has been estimated according to response surface methodology. As shown in Figure 8, when the stripping time is 90 min, the pH value is 11 and the gas flow rate is 28 m^3^/h, the maximum value for the corresponding response value is 78.13%. As shown in Figure 9, the reliability of the satisfaction function of the optimized combined parameters is 0.153, indicating that the optimized function can represent the test model and required conditions [11].

To verify the reliability and applicability of the regression model, the optimized combined parameters predicted by response surface methodology are adopted for verification tests. The results of three tests are shown in Table 5. The average removal rate of ammonia nitrogen from biogas slurry is 73.93%, which is only 4.2% different from the predicted value, indicating that the test value and predicted value are quite close and further verifying that the regression model provides accurate and effective analysis of the interactions among factors influencing the ammonia stripping effect of biogas slurry and has optimized the prediction of combined parameters.

Jin optimized the conditions for ammonia stripping of anaerobic effluent from dairy farming wastewater through Box–Behnken design via response surface tests [11]. The optimized combined parameters were as follows: stripping time of 5.3 h, pH value of 11.5, and temperature of 32.5 °C; the predicted maximum ammonia nitrogen removal rate was 92.8%. Hossini et al. adopted response surface methodology to optimize parameters including ammonia nitrogen quality concentration, pH value and temperature [27]. The optimal technological conditions were as follows: ammonia nitrogen concentration of 1440 mg/L, pH value of 10.7 and temperature of 36 °C. Under the optimal process conditions, the system was operated at a gas flow rate of 3 L/min for 12 h, and the maximum removal rate of ammonia nitrogen was 84.11%. Huang et al. used honeycomb packing with a controlled stripping time of 120 min and utilized response surface methodology to optimize the stripping method for pharmaceutical wastewater with a high ammonia nitrogen concentration [28]. The optimal combined technological conditions were as follows: pH value of 11.5, gas–liquid ratio of 3126, and temperature of 50.5 °C; the predicted maximum ammonia nitrogen removal rate was 92.16%. Compared with ammonia stripping technology studies conducted by other scholars, the combined parameters obtained in this experiment can achieve a higher stripping efficiency within a shorter stripping time; furthermore, consideration is given to both ammonia nitrogen removal rate and economic efficiency.

## 4. Conclusions

Based on the study, a hollow polyhedral packing with an optimized stripping time of 90 min, pH value of 11, gas flow rate of 28 m^3^/h, gas–liquid ratio of 2000, and temperature of 30 °C, produced an average ammonia nitrogen removal rate of 73% in biogas slurry. The technological parameters obtained can provide a reference for the optimization of ammonia stripping technology in practical application.

## Figures and Tables

**Figure 1 ijerph-16-03819-f001:**
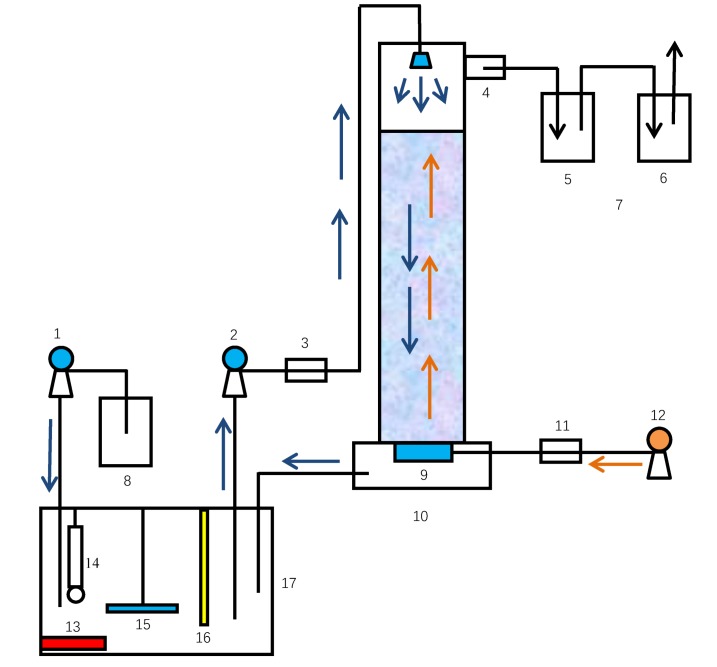
Schematic diagram of stripping components. 1. Alkaline solution peristaltic pump; 2. Biogas slurry peristaltic pump; 3. Liquid flowmeter; 4. Gas outlet; 5. Absorption tank; 6. Neutralization tank; 7. Absorption plant; 8. Alkaline solution tank; 9. Aeration device; 10. Stripping tower; 11. Gas flowmeter; 12. Aeration fan; 13. Heating rod; 14. pH meter; 15. Stirrer; 16. Temperature thermocouple; 17. Liquid storage tank.

**Figure 2 ijerph-16-03819-f002:**
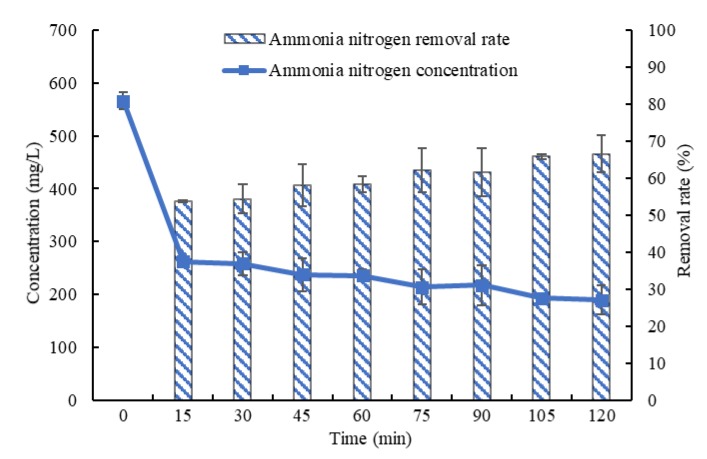
Variations in the concentration and removal rate of ammonia nitrogen with time in biogas slurry.

**Figure 3 ijerph-16-03819-f003:**
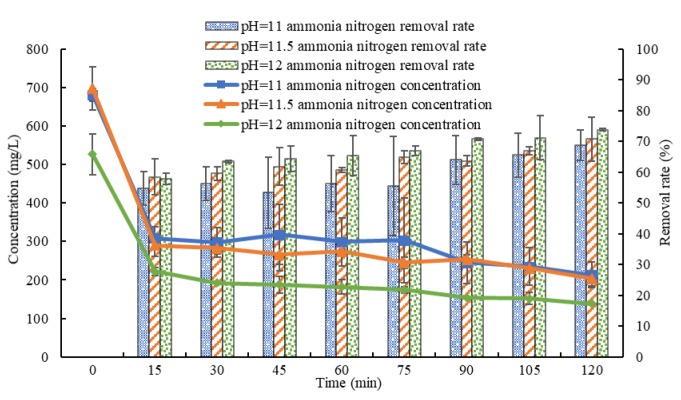
Variations in the concentration and removal rate of ammonia nitrogen with time in biogas slurry with different pH values.

**Figure 4 ijerph-16-03819-f004:**
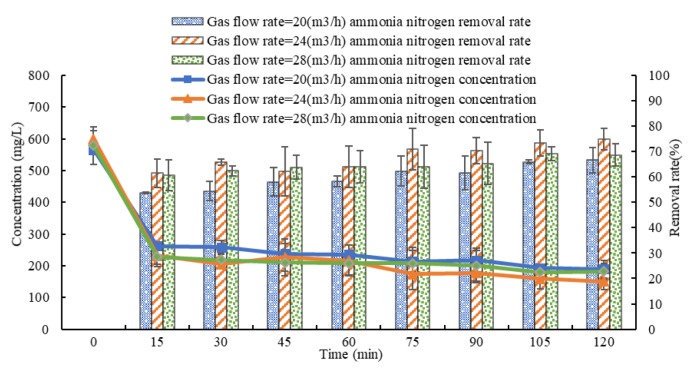
Variations in the concentration and removal rate of ammonia nitrogen with time in biogas slurry with different gas flow rates.

**Figure 5 ijerph-16-03819-f005:**
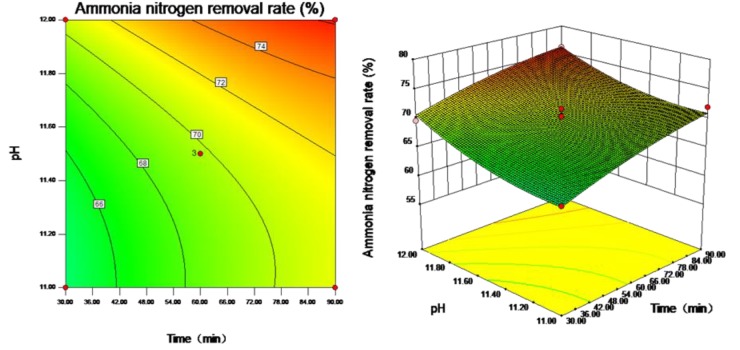
Contour plot and response surface plot of the effect of stripping time and pH value on ammonia nitrogen removal rate.

**Figure 6 ijerph-16-03819-f006:**
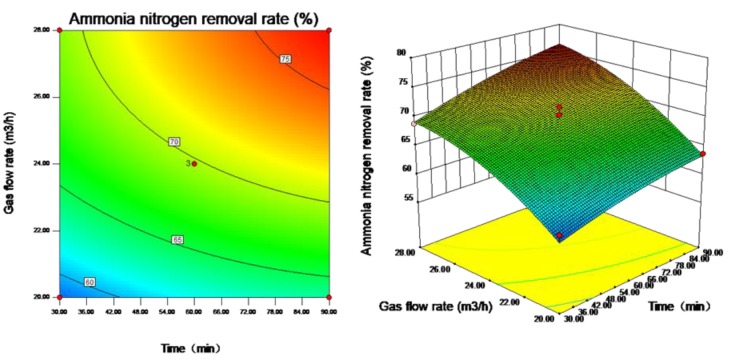
Contour plot and response surface plot of the effect of stripping time and gas flow rate on ammonia nitrogen removal rate.

**Figure 7 ijerph-16-03819-f007:**
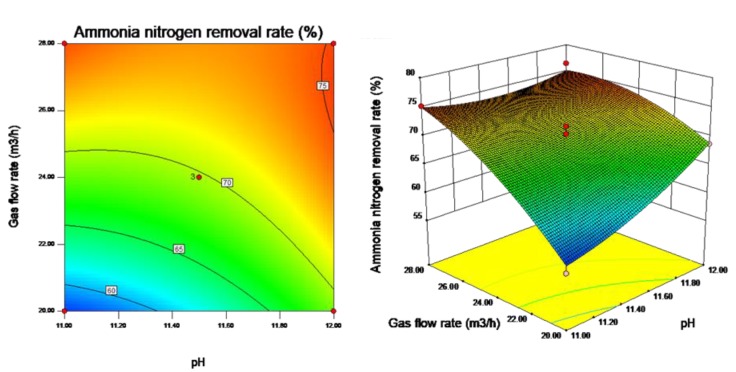
Contour plot and response surface plot of the effect of pH value and gas flow rate on ammonia nitrogen removal rate.

**Figure 8 ijerph-16-03819-f008:**
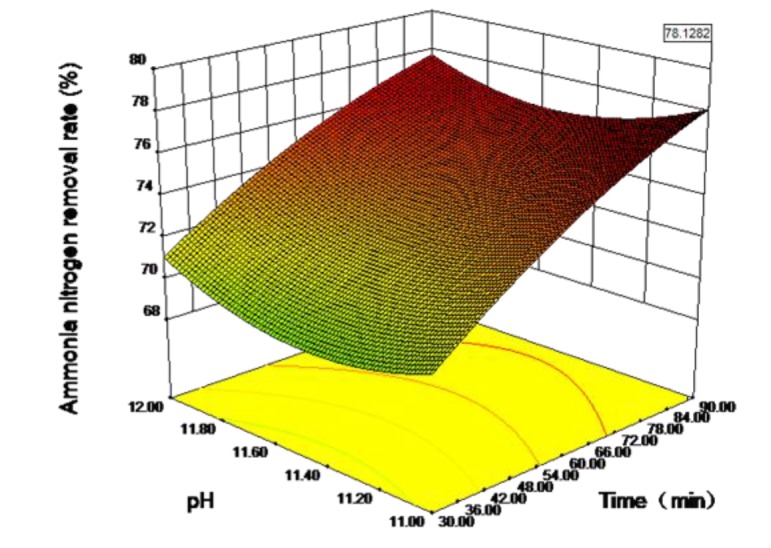
Response surface plot of the predicted value of ammonia nitrogen removal rate with different pH values and stripping time.

**Figure 9 ijerph-16-03819-f009:**
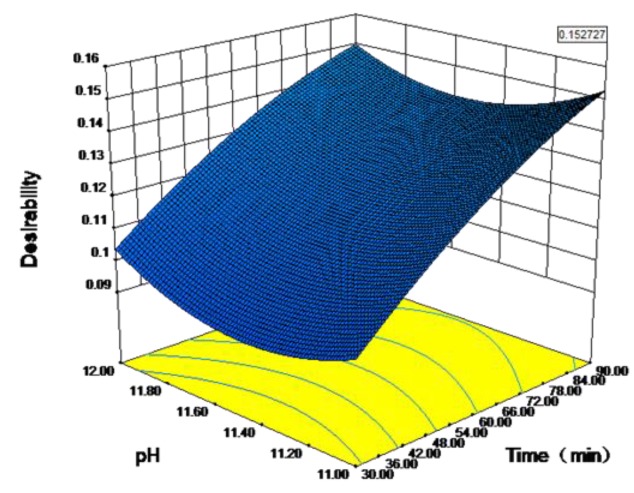
Response surface plot of the desirability function with different pH values and stripping time.

**Table 1 ijerph-16-03819-t001:** Scheme of single-factor ammonia stripping experiment.

Parameter	Control		
	Time (min)	pH	Gas Flow Rate (m^3^/h)
Time (min)	15, 30, 45, 60, 75, 90, 105, 120	10.5	20
pH	120	11, 11.5, 12	20
Gas flow rate (m^3^/h)	120	10.5	20, 24, 28

**Table 2 ijerph-16-03819-t002:** Factors and levels of response surface experiment.

Level	Factor A Time (min)	Factor B pH	Factor C Gas Flow Rate (m^3^/h)
−1	30	11	20
0	60	11.5	24
1	90	12	28

**Table 3 ijerph-16-03819-t003:** Schemes and results of the response surface experiment.

Number	Factor A	Factor B	Factor C	Response Y
	Time (min)	pH	Gas Flow Rate (m^3^/h)	Ammonia Nitrogen Removal Rate (%)
1	30(−1)	11(−1)	24(0)	64.40
2	90(1)	11(−1)	24(0)	71.99
3	30(−1)	12(1)	24(0)	69.65
4	90(1)	12(1)	24(0)	76.06
5	30(−1)	11.5(0)	20(−1)	59.52
6	90(1)	11.5(0)	20(−1)	63.64
7	30(−1)	11.5(0)	28(1)	68.85
8	90(1)	11.5(0)	28(1)	75.21
9	60(0)	11(−1)	20(−1)	56.16
10	60(0)	12(1)	20(−1)	68.65
11	60(0)	11(−1)	28(1)	75.16
12	60(0)	12(1)	28(1)	76.48
13	60(0)	11.5(0)	24(0)	71.71
14	60(0)	11.5(0)	24(0)	70.39
15	60(0)	11.5(0)	24(0)	67.11

**Table 4 ijerph-16-03819-t004:** Analysis of variance of quadratic polynomial regression equation.

	SS	df	MS	F	P	Significance
Model	489.60	9	54.40	13.82	0.0050	**
A	74.91	1	74.91	19.03	0.0073	**
B	66.87	1	66.87	16.99	0.0092	**
C	284.77	1	284.77	72.33	0.0004	**
AB	0.35	1	0.35	0.088	0.7782	
AC	1.25	1	1.25	0.32	0.5968	
BC	31.19	1	31.19	7.92	0.0373	*
A^2^	2.13	1	2.13	0.54	0.4950	
B^2^	8.85	1	8.85	2.25	0.1941	
C^2^	17.42	1	17.42	4.42	0.0894	
Residual	19.68	5	3.94			
Lack of Fit	8.46	3	2.82	0.50	0.7180	
Pure Error	11.22	2	5.61			
Cor Total	509.29	14				

Comment: * significant difference (*p <* 0.05), ** extremely significant difference (*p <* 0.01).

**Table 5 ijerph-16-03819-t005:** Conditions and results of the verification experiment.

Number	Time (min)	pH	Gas Flow Rate (m^3^/h)	Ammonia Nitrogen Removal Rate (%)
1	90	11	28	73.74
2	90	11	28	75.17
3	90	11	28	72.88

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
