# Peer review of "Optimization of Ammonia Stripping of Piggery Biogas Slurry by Response Surface Methodology"

_ijerph, 2019, doi:10.3390/ijerph16203819_

Round 1

Reviewer 1 Report

1. Quality of Figures 5, 6,7, 8 & 9 should be improved, I can not  read the values of X-axis and Y-Axis.

2. spelling error, page 2, line 75.

"After the air and 74 stripping offgas are discharged form the exhaust port, ammonia gas is absorbed by the absorption.."

 "...form..."-   Should it  be  replaced by "...from...."  ?  

Author Response

Response 1: Dear reviewer, I have remade and enlarged the figures 5,6,7,8 & 9 to see it clearly for readers.

Response 2: I have corrected the spelling error the reviewer pointed out.

Reviewer 2 Report

Title:   Optimization of Ammonia Stripping of Piggery 2 Biogas Slurry by Response Surface Methodology

Authors: Mengyuan Zou, Hongmin Dong, Zhiping Zhu and Yuanhang Zhan

Manuscript ID: IJERPH-605605

My comments:

This research article optimized the operating parameters of ammonia stripping system of piggery biogas slurry following response surface methodology. Given the continued interest of the general public in ammonia stripping and recovery, the study is timely and significant. Additionally, the study appears to be well-planned. Truly, this paper is a well-written, grammar is good, easy to follow, showed sequential flow of information, and the authors have demonstrated good knowledge of the overall problem. I also believe the authors have competently made the described measurements in the laboratory. Per my opinion, this paper would be published with minor justifications/corrections as suggested below:

Why narrow ranges of pH and gas flow were evaluated instead of using wide range, which could have display better discernable effects on ammonia stripping.

Authors are strongly advised to calculate and use RMSE (root means square error) instead of CV for model comparison. Generally, higher R2 with lower RMSE is considered better model.

Authors are strongly advised to rewrite conclusion. A conclusion should include only the solid findings instead of rewriting the methods and objectives. My suggestion is given below:

Conclusion:

A hollow polyhedral packing with an optimized stripping time of 90 min, pH value of 11, gas flow rate of 28 m3/h, gas-liquid ratio of 2000, and temperature of 30℃, produced an average ammonia nitrogen removal rate of 73% in biogas slurry.

Author Response

Response 1: Dear reviewer, the pH value and gas flow rate were evaluated with narrow range because this test is based on the previous results of our research group. It is stated in the text as follows:

"In our research group, Sui (2014) showed that the removal rate of ammonia nitrogen in biogas slurry was 81.84% under the operating conditions of pH 10.5, gas flow rate 5 m3/h, gas-liquid ratio 2000-2500 and temperature 30 °C. However, the stripping time is more than 12 hours. Therefore, this test is based on the parameters of pH 10.5, gas-liquid ratio 2000 and temperature 30 °C. The pH value and gas flow rate are appropriately increased."

Response 2: RMSE has been calculated and used instead of CV for model comparison in text.

"The multiple linear regression equation relating the ammonia nitrogen removal rate (Y) to the stripping time (A), pH value (B) and gas flow rate (C) is Y=69.00+3.06A+2.89B+5.97C, the determination coefficient is R2=0.8376, the adjusted determination coefficient is R2=0.7932, and the root mean square error (RMSE) is 11.92;

the quadratic polynomial regression equation relating the ammonia nitrogen removal rate (Y) to the stripping time (A), pH value (B) and gas flow rate (C) is Y=69.74+3.06A+2.89B+5.97C-0.29AB+0.56AC-2.79BC-0.76A2+1.55B2-2.17C2, the determination coefficient is R2=0.9613, the adjusted determination coefficient is R2=0.8918, and the RMSE is 7.38."

"The root mean square error (RMSE) indicates the degree of dispersion of the sample, the smaller the RMSE, the more reliable the model. "

Response 3: The conclusion has been rewritten with reference to the reviewer's suggestion as follows.

"Based on the study, a hollow polyhedral packing with an optimized stripping time of 90 min, pH value of 11, gas flow rate of 28 m3/h, gas-liquid ratio of 2000, and temperature of 30℃, produced an average ammonia nitrogen removal rate of 73% in biogas slurry. The technological parameters obtained can provide a reference for the optimization of ammonia stripping technology in practical application."

Reviewer 3 Report

Dear Authors,

Thank you very much for the possibility of reading your article. 

The ammonia stripping method is a relatively simply and effective process removing  of ammonia from different waste water that's why it has emerged as a strong interest research area among researchers and industrial. 

In this paper response surface method was used to optimize conditions (pH value, gas flow rate, stripping time) the ammonia stripping of piggery biogas slurry. Using response surface method effectively reduces cost of test and gives the possibility to obtain reliable results. The highest reduction of ammonia concentration (73%) under optimal economic efficiency was obtained at a gas flow rate 28 m3/h, pH value of 11 and temperature of 30°C. These results show the possibility of achieving high ammonia reduction in the relatively short time. 

General comments:

The same record of units should be using, e.g. mg/L or mg L-1.

The methodology should be completed:

-what was absorption liquid

-what was the concentration of the alkalizing solution

-how the ammonia concentration was measured

Specific comment:

Line 93: there is PH meter - there should be pH meter.

Author Response

Response 1: Dear reviewer, I have unified the units including m3/h and mg/L in text and figures.

Response 2: The methodology has been completed as follows.

"The absorption liquid used in the test is 1 mol/L sulfuric acid solution. The neutralizing liquid used in the test is 0.5 mol/L sodium hydroxide solution."

"The ammonia nitrogen concentration is measured by salicylic acid-sodium hypochlorite spectrophotometry. The test instrument is HACH COD Reactor Model DR 6000 (HACH Company, USA). The average ammonia nitrogen removal rate (Y) is calculated by the following equation (in text):

where C is the ammonia nitrogen concentration in the test water before stripping and C' is the ammonia nitrogen concentration in the test water after stripping."

Response 3: I have corrected the spelling error the reviewer pointed out.